# Effect of Using Mobile Phones on Driver’s Control Behavior Based on Naturalistic Driving Data

**DOI:** 10.3390/ijerph16081464

**Published:** 2019-04-25

**Authors:** Lanfang Zhang, Boyu Cui, Minhao Yang, Feng Guo, Junhua Wang

**Affiliations:** 1Key Laboratory of Road and Traffic Engineering of the Ministry of Education, Tongji University, Shanghai 201804, China; zlf2276@tongji.edu.cn (L.Z.); 1733299@tongji.edu.cn (B.C.); 2Shanghai Urban Construction Design and Research Institute, Shanghai 200125, China; yangminhao@sucdri.com; 3Department of Statistics, Virginia Polytechnic and State University, Blacksburg, VA 24061, USA; feng.guo@vt.edu

**Keywords:** natural driving data, distracted driving, mobile phone, moving time window, driving control behavior

## Abstract

Distracted driving behaviors are closely related to crash risk, with the use of mobile phones during driving being one of the leading causes of accidents. This paper attempts to investigate the impact of cell phone use while driving on drivers’ control behaviors. Given the limitation of driving simulators in an unnatural setting, a sample of 134 cases related to cell phone use during driving were extracted from Shanghai naturalistic driving study data, which provided massive unobtrusive data to observe actual driving process. The process of using mobile phones was categorized into five operations, including dialing, answering, talking and listening, hanging up, and viewing information. Based on the concept of moving time window, the variation of the intensity of control activity, the sensitivity of control operation, and the stability of control state in each operation were analyzed. The empirical results show strong correlation between distracted operations and driving control behavior. The findings contribute to a better understanding of drivers’ natural behavior changes with using mobiles, and can provide useful information for transport safety management.

## 1. Introduction

Related studies have shown that using mobile phones during driving is one of the leading causes of traffic accidents [1,2,3,4,5,6,7,8]. In response to this problem, nearly 70 countries and regions have enacted laws and regulations banning the use of mobile phones during driving [9]. However, most countries only prohibit the making or receiving of hand-held calls, and there is no prohibition on the use of hands-free devices [10]. Differences in regulations indicate that there is still a lack of unified understanding of the impact of using mobile phones on driving behavior. At the same time, these regulations haven’t been widely accepted by drivers [11,12,13,14], and this distracted behavior is becoming more frequent with the explosive development of smartphone functions.

Driving tasks mainly include controlling the stability of vehicles and monitoring the driving environment, which are directly related to driver’s manual operation and visual attention. Some previous studies tend to take visual distraction behavior as the main factor of accident. For example, the mean percentage of “total eyes-off-road time” (TEORT) is 33.1% and 59.5% when the driver performs answering and dialing operations on a hand-held call. However, the percentage of TEORT is 9.5% or 15.6% when the driver communicates by phone in the hand-held or hands-free mode [15,16]. In addition, most operations of using mobile phones are manual–visual distractions, such as texting and dialing. The effect of manual–visual distractions on driving performance is negative and of greater impact than just visual distraction [17,18,19]. Therefore, many countries or regions, when formulating relevant laws and regulations, believe that manual–visual interference in handheld phone calls will have a significant impact on driving safety, while ignoring the impact of cognitive interference on driving behavior. However, the cognitive distraction caused by using mobile phones will affect the performance of driving tasks from two aspects according to ergonomics theory. Cognitive psychology holds that driver’s attention during driving is limited [20]. Driver’s attention typically splits between performing control operations and processing driving environment information. When the distraction demand caused by mobile phone use is high, a part of attention is required to be allocated to deal with the distracted operation, thus causing interference to the driver’s control behavior and environmental monitoring ability. According to Wickens’ Multiple Resource theory [21], although conversation is auditory and driving tasks are visual, cognitive distraction interferes with visual behavior as the conversation triggers driver’s visual memory. As a result, drivers’ ability to process visual information becomes more sluggish, and less sensitive to performing control operations and monitoring the driving environment. Collet’s study compared changes in driver’s heart rates and skin resistance during hand-held calls and conversations with passengers. The results show that when drivers engage in these two kinds of distracted behaviors, their physiological indicators are significantly changed and there is no significant difference as to the degree of influence [22].

Early studies on distracted driving related to mobile phone use were mainly based on using driving simulators to explore the impact on driving behavior. It was found that hand-held calls or sending and receiving texting messages during driving caused a significant decrease in speed, making it difficult to maintain stable speed and headway distance [23,24,25]. In the lateral direction, the lateral deviation increased, the frequency of lane changes and lateral stability decreased [26,27]. At the same time, drivers’ ability to recognize signals was compromised and their reaction time lengthened [28,29,30]. The driving simulation environment provides the opportunity of data acquisition for related research, and has achieved a lot of research results. However, the driving simulation research is experimental and participants are often required to perform distraction behavior in a relatively unfamiliar driving environment. Participants couldn’t choose whether to perform distracted behaviors according to the driving state and environment, which is quite different from real situations and leads to certain limitations of results.

With the launch of large-scale natural driving projects, such as the Second Strategic Highwasy Research Plan Naturalistic Driving Study (SHRP2 NDS), UTDrive, and Shanghai NDS (SH-NDS), researchers began to conduct mobile phone distraction study based on natural driving data [7,8,10,15,31,32,33,34]. Natural driving research reduces the interference of experimental arrangements on driving behavior and provides the opportunity to observe the actual driving process with unobtrusive high precision data [35]. However, unlike a lab-controlled experiment, many more factors of the driving environment should be taken into consideration. Most research didn’t consider the interference of driving factors such as road scenes and traffic conditions, which affected the results. In addition, the above studies generally use the mean or instantaneous values of parameters such as speed, distance, lane offset or steering wheel angle when constructing indexes to represent the running state of vehicles. In the state of natural driving, the driver will make some self-adjustment when the distraction is weak, so that these indicators may not be significantly affected, and it is difficult to fully represent the performance of the driver and changes of driving state in the whole process. In addition, the process of using mobile phones for hand-held calls involves multiple operations such as answering, dialing, talking, and hanging up. The distractions caused by different operations differ in their type, intensity and duration. In previous studies, the influence of mobile phone use on driving performance and hand-held call behavior is often regarded as a whole process without distinguishing and comparing different operations according to the characteristic of distraction. In summary, the understanding of the influence of using mobile phones on driving behaviors, especially drivers’ control behaviors, under natural driving conditions is still vague.

Based on the natural driving data of Shanghai NDS, this paper attempts to explore the influence on driver’s control behavior of five operating behaviors of using mobile phones in a natural driving state: answering, dialing, talking and listening, hanging up and viewing information. We collected samples based on pre-defined screening conditions to reduce the influence of driving environment factors, and categorize the different operations of mobile phone use into five types. Based on driver’s distraction mechanism, multiple metrics representing the driver’s control behavior are constructed using moving time window, including the index of control activity’s intensity, control operation’s sensitivity, and control state’s stability.

## 2. Methods

### 2.1. Shanghai Natural Driving Study

The Shanghai Natural Driving Research Project (SH-NDS), jointly conducted by the Tongji university, the General Motors, and Virginia Tech Transportation Institute, is the first study on natural driving in China. The project collected the driver’s behavior for an extended time period (up to two months), thus minimizing the interference to the driver’s daily driving behavior. A total of 60 drivers, aged between 35 and 50, participated in the study, and each driver had more than five years of prior driving experience. The experimental vehicles included five GM branded vehicles with automatic transmission vehicles. To date, six phases of the project have been launched, with more than 750,000 km of driving data collected. As shown in Table 1 and Figure 1, the data acquisition system (DAS) consists of Doppler radar, Triaxial accelerometer, GPS and four synchronous cameras. The DAS can continuously collect driving state information, driver behavior information and external environment information simultaneously. The stored data can be divided into numerical data and video data.

### 2.2. Sampling

Under natural driving conditions, changes in road environment and traffic conditions have a great impact on driving control behavior. In order to reduce the external factors that may interfere with the analysis results, five sample screening conditions were set in the sample acquisition process, listed below.
The vehicle was driving on an expressway or freeway when the event occurred.;The vehicle was in continuous traffic flow.No other events occurred at the same time.No obvious interference from surrounding vehicles (no emergency brake or sharp turn) was detected.There was a period of normal driving before and after the event to get control samples.

The distracted driving sample was screened by checking the driver’s facial video and hand video. The beginning of a cellphone use event is defined as when the driver’s eye-glance shifted and a hand began to reach for the phone. The event is considered to be over when the driver’s eye-glance returned to driving tasks and hands were back on the steering wheel. To meet the screening conditions, the surrounding driving environment condition for each cellphone use event was examined by checking the front video and the rear video. Moreover, video data and numerical data in the Shanghai natural driving project adopt a unified timeline, and the corresponding bus data can be extracted through the timestamp in the video.

### 2.3. Cellphone Subtasks

The use of mobile phones during driving can be classified into two types: the hand-held call behavior and the information viewing behavior. Among them, hand-held call behavior is further categorized into answering calls and making calls. As shown in Table 2, the hand-held call behavior includes four sub-tasks: answering or dialing, talking and listening, and hanging up. Amongst these sub-tasks, the visual distraction is reduced subsequently into viewing, dialing, answering and hanging up. In addition, answering, dialing and viewing create conspicuous manual distraction. The communication process mainly leads to cognitive distraction of the driver. Table 2 lists the criteria for the occurrence of each operation, and the occurrence of the next operation or normal process determined the end of the previous operation. This paper obtained the numerical driving data in the corresponding operation process by recording the time stamp in the video when the operation occurs and ends. In addition, as shown in Figure 2, the normal driving process of 3 s before and after the use of mobile phone was taken as the control case. At the same time, in order to avoid the influence of using mobile phone on the control sample, a 1 s interval was set between the normal process interval and the distraction interval.

### 2.4. Data Analysis

In the process of using mobile phones, drivers will reallocate their attention resources to cope with the workload of distracted operations on driving tasks, which will affect the intensity of relevant control activities. At the same time, distracted operation will make the driver more sluggish in acquiring and processing environmental information, and thus the sensitivity of driving control operation will be partially compromised. Under the influence of the above two effects, the stability of driving control state will likely to fluctuate accordingly. Therefore, this paper examines the driving control behavior based on the dynamic data collected in the natural driving project, and analyzes the changes of the intensity of driver’s control activities, the sensitivity of control operation, and the stability of control state under different operations. In the process of using mobile phones, there is a high degree of distraction in specific moments, which has a significant impact on the driver’s control behavior. In order to explore this effect, it is necessary to make full use of the sequence data of the natural driving study to construct indexes that can obtain the running characteristics of vehicles in the process of using mobile phones.

In this paper, the concept of moving time window was used to construct the characteristic index of driver’s control behavior [36]. As shown in Figure 3 α represents the parameter value, such as velocity and acceleration, and the standard deviation of α was used as a measure of fluctuation in a 1 s time window to detect the minimum variation of the parameter. The collection frequency of dynamic data in SH-NDS is generally 10 Hz, so a 1 s time window can be built every 0.1 s. In the process of a sub-operation, the standard deviation of parameters within each 1 s time window was calculated, and the standard deviation of the $i^{th}$ window is denoted by $STD_{i}$, forming a new sequence composed of standard deviations.

In the state of natural driving, vehicle kinematic parameters such as speed, acceleration and lane deviation are sensitive to road traffic condition and the driver’s own driving intention. However, when the driver’s control ability remains in a normal state and is not strongly affected by the driving environment (such as sharp turning, traffic congestion, etc.), the fluctuation of these kinematic parameters within each second will likely to remain within a certain range without drastic changes. The fluctuation of parameter values will be smaller in a short period under normal conditions. Therefore, under the premise of no drastic changes under normal driving environment, this stability may be lost when the driver’s control ability is affected significantly. Therefore, the one-second standard deviation value calculated in each time window reduces the interference of other factors and can be used to characterize the influence of distracted operation. In addition, the influence of distracted operation will vary with the degree of distraction in the process of using a mobile phone. This processing method can make full use of the sequence data for preliminary feature extraction. The sequence of standard deviation values obtained by preliminary calculation can ensure that sufficient data has a strong correlation with the distracted operation and facilitate the subsequent construction of feature indicators.

#### 2.4.1. Control Activity Intensity

The intensity of the control activity reflects the frequency and enthusiasm of the driver in the control operation and represents the attention resource allocation of the driver in the execution of the primary driving tasks. The acceleration of a vehicle is a parameter directly related to driving control operations, such as controlling accelerator or brake pedals, the steering wheel, etc. Therefore, longitudinal acceleration and lateral acceleration are adopted in this paper to construct characteristic indexes representing the driver’s control activities. The standard deviation of the acceleration value in a 1 s window reflects the strength of the driver’s control activities in that second. In this paper, the median value of the standard deviation sequence of the acceleration is selected to represent the control activity of the driver in each process.

#### 2.4.2. Control Operation Sensitivity

The sensitivity of the control operation is mainly related to the driver’s awareness of road information and the execution of the control operation. The change of control operation needs to correspond with the change of driving environment in time. When the degree of driver distraction is high, the sensitivity of executing control operation begins to decline. In this paper, the standard deviation sequence of lateral and longitudinal acceleration values during each operation was calculated. In the $I^{th}$ time window, the $STD_{i}$ values of the longitudinal and lateral accelerations were constructed as two-dimensional vectors to represent the overall control activities of the driver in the $i^{th}$ window. If the driver’s cognitive delay occurs, the time to change from one control activity state to another state will increase, which is manifested as a significant decrease in the sensitivity of the control operation within a certain time range. Therefore, the standard Euclidean distance (*d*) of vectors of adjacent windows was further calculated to represent the change of the driver’s control activity at the adjacent moment, and the median value of all the distance values in a process was used to characterize the sensitivity of the driver’s control operation change in that process.
(1)$$d=\sqrt{\sum_{j=1}^{2}\frac{{\left(SD_{\left(i\right),j}-SD_{\left(i+1\right),j}\right)}^{2}}{{S_{j}}^{2}}}$$
where: i  is the sequence number of the moving time window; the value of *j* is {1,2}, representing two types of acceleration; $SD_{\left(i\right),j}$ is the standard deviation of the *j* acceleration value in the $i^{th}$ window; *Sj* represents the standard deviation of the sequence of the standard deviation of the *j* acceleration during the whole process; *d* is the calculated value of the standard Euclidean distance of the adjacent points.

#### 2.4.3. Control State Stability

Speed and lane offset are direct driving performance of vehicles in two directions, which have a strong correlation with driving safety. Therefore, this paper used the values of speed and lane offset to construct indexes to represent the stability of longitudinal and lateral control state of drivers to evaluate the stability of control states. As shown in Figure 4,$\;S_{j}$ is the standard deviation of the standard deviation sequence of speed values during the whole process, which was used to characterize the fluctuation of the longitudinal driving control state of the process. Similarly, the $S_{j}$ value of lane offset was used to represent the stability of the driver’s lateral control state during the operation. The $S_{j}$ represents the fluctuation of the control state of a certain parameter in a process, not only the fluctuation of the parameter, so it can be used to evaluate the stability of the control state of the whole process.

## 3. Results

As shown in Table 3, according to the sample screening conditions, a sample of 134 cases of five kinds of distracted operations involving 25 drivers are extracted in this paper to analyze the influence of different operations on driving control behaviors.

### 3.1. Answering the Call

During the answering process, the intensity of the driver’s longitudinal control activity was decreased significantly (T(59) = 2.659; *p* = 0.010), but no significant changes were found in other sub-operations. As shown in Table 4, Figure 5 and Figure 6, the sub-operations of answering, talking and listening, and hanging up all led to a significant decrease in the sensitivity of the driver’s control operation. In the process of answering and talking, the stability of the longitudinal control state was significantly affected, and the influence of talking and listening was significantly higher than other distracted operations. In addition, the statistical results show that there was no significant influence of the call answering behavior on the driver’s lateral control activities (F(1.6, 94.8) = 3.033, *p* = 0.064) and lateral control state (F(2.9, 78.6) = 0.668, *p* = 0.570)).

### 3.2. Dialing the Call

The statistical results show that compared with the normal driving process, the dialing operation resulted in a significant decline of driver’s longitudinal control activity (T(38) = 2.045, *p* = 0.048), a significant decline in the sensitivity of the control operation (T(38) = 2.315, *p* = 0.026), and a deterioration in the stability of the longitudinal control state (T(38) = −4.466, *p* < 0.01). However, no significant change was found in the lateral control behavior (T(38) = 1.385, *p* = 0.174; T(38) =−0.806, *p* = 0.425).

### 3.3. Viewing Information

In terms of control activities, the longitudinal control activity in the process of viewing information weren’t affected (T(34) = 1.839, *p* = 0.075; T(34) = −0.617, *p* = 0.514), but the intensity of lateral control activity was significantly decreased (T(34) = 2.498, *p* = 0.017; T(34) = −2.044, *p* = 0.049).

In terms of control sensitivity, driver’s sensitivity of control operation in the process of viewing was significantly lower than that before distraction. The results show that there were significant differences between the control samples before and after the distraction. Therefore, this paper further selected four segments of 2–5 s (N_3(2s)_), 3–6 s (N_3(3s)_), 4–7 s (N_3(4s)_) and 5–8 s (N_3(5s)_) after the end of distraction as control samples for comparison. As shown in Figure 7 and Table 5, the sensitivity of the driver’s control operation returned to the normal state after the distraction behavior ended for 5 s.

Eventually, the operation of viewing resulted in a deterioration of the longitudinal driving control state (T(34) = −3.265, *p* = 0.002; T(34) = 5.709, *p* < 0.01), while the lateral control state was not significantly affected (F(1.9, 63.2) = 1.429, *p* = 0.247).

## 4. Discussion

In the process of using mobile phones, answering, dialing and viewing are the most complex manual–visual distraction operations. The intensity of the driver’s longitudinal control activity is decreased during answering and dialing, while the lateral control activities remain normal. In contrast, the intensity of lateral control activity is decreased in the process of viewing, while the longitudinal control activities aren’t significantly affected. The results indicate that drivers have different self-regulatory strategies in different distractions.

Some studies show that distracted drivers often exhibit self-regulatory behaviors, such as reducing driving speed, increasing the distance to the lead vehicle, and making less lane changes, which are considered to reduce the driving demand and accident risk [37,38,39]. Some studies have also found less lane deviation during hand-held calls and increased lane deviation during texting [40,41,42,43], which is consistent with the results of this paper. Also, some studies report dissimilar results [44,45], especially in natural driving studies [33,34]. This may be related to the complexity of natural driving samples. At the beginning of this study, some interfering factors are eliminated by setting sample screening conditions and index construction. The intensity of the control activity can reflect the decision-making willingness of drivers. The results show that drivers are unable to devote more attention resources in the face of extra workload caused by complex manual–visual distractions, and tend to reduce longitudinal control activities, so as to ensure that the intensity of lateral control activity is not affected. As the distraction intensifies, the strategy shifts. Although this behavior is a kind of self-protection of the driver, the longitudinal self-regulatory behaviors of drivers have a significant negative impact on the surrounding traffic flow [46]. Under the condition of high-speed driving, the influence of this regulation on driving risk in the region needs to be further explored.

In addition, during the process of answering and dialing, the sensitivity of the driver to perform control operations is significantly affected, which proves that the distracted operation leads to a duller response of the driver. From the perspective of the stability of the control state, under the influence of the above two aspects, the stability of the driver’s longitudinal control state during the answering and dialing process is deteriorated, but the stability of the lateral control state can still be maintained in a normal state.

Hang up operation has little influence on the driver’s control behavior. The analysis results show that although the operation weakens the sensitivity of the driver’s control operation, it doesn’t have a significant impact on the stability of vehicle control, indicating that the driver can cope with the distraction caused by this operation.

The analysis results of this paper show that the talking and listening operation has no significant impact on the intensity of the driver’s control activities, indicating that the distraction exerts less workload on the driver. However, the effect of cognitive distraction caused by talking and listening on the driver’s control sensitivity is similar to that of dialing and answering. Moreover, the influence of talking and listening operation on the stability of the longitudinal control state of the driver is more severe, indicating that the cognitive distraction will also have a significant impact on the stability of driving. Compared with other operation processes, the duration distribution of talking and listening process is longer and more discrete, while the distraction duration may affect the vehicle control stability to some extent. Therefore, 81 talking and listening operation samples satisfying the screening conditions (average duration = 50.1 s, variance = 46.1) are extracted to further explore the correlation between duration and the stability of longitudinal control state. As shown in Table 6, no significant correlation has been found within the duration range of the current study.

In the process of viewing information, viewing also leads to a significant decrease in the sensitivity of the driver’s integrated control operation. Moreover, different from other operations, the effect of cognitive distraction caused by the driver’s brain processing information still exists 4–5 s after viewing operation. According to the analysis results of the control state, although the driver’s lateral control activities are significantly affected, the stability of the lateral control state is not affected, while the longitudinal control state shows significant fluctuations. Therefore, this paper argues that maintaining longitudinal stability requires more effort from the driver than maintaining lateral stability, which makes the stability of longitudinal control more susceptible to influence.

## 5. Conclusions

Based on the Shanghai natural driving project, this paper collected high-precision sample data of distracted driving with mobile phones. Compared with driving simulation experiments and real car experiments, natural driving study provides data with higher quality and authenticity. In the sample acquisition, several screening conditions are set to reduce the influence of confounding factors, so as to increase the reliability of the analysis results. At the same time, based on the theory of ergonomics, this paper argues that the use of mobile phones mainly impacts the intensity of the driver control activity, the sensitivity of control operation and the stability of vehicle control state. Furthermore, based on the concept of moving time window, we make full use of the natural driving sequence data to construct the characteristic index. Finally, according to the behavior characteristics of different stages in the process of using mobile phone, this paper classifies the distraction process into different sub-operations and analyses the influence on driver’s control behavior. The conclusions are as follows.

The study finds that drivers used different resource allocation strategies to deal with the extra workload caused by distraction under different distraction operations. Typical manual–visual distraction operations, such as answering, dialing and viewing, can result in a significant interference in the driver’s attention resources. Therefore, the driver has adopted some coping strategies under different distraction intensity. The driver tends to decrease the longitudinal control activities in the process of answering or dialing, and decrease the lateral control activities in the process of viewing, which plays an important role in reducing driving demand.

In the process of hand-held calls, talking and listening leads to a decrease in the sensitivity of the driver’s control operation, and the degree of influence is no different from other manual–visual operations. In addition, talking and listening leads to the most significant reduction in the stability of longitudinal control state in the process of answering calls. The influence of talking and listening operation on driver’s control behavior is no less than that of other operations, indicating that banning only the handheld communication mode cannot eliminate the security risks caused by the cognitive distraction. The use of hands-free calls should be restricted. Natural driving research for hands-free calling will be conducted in subsequent studies.

Nowadays, with the development of the functions of smart phones, the behavior of using mobile phones to edit short messages is less frequent, while the behaviors of browsing and viewing is more frequent. The advent of social media, mobile navigation and especially ride-hailing services has left drivers completely dependent on their phones. However, with the proliferation of this behavior, there is a lack of effective regulation in China. It can be seen that the viewing behavior not only has a significant influence on the sensitivity and stability of driving control, but also that the influence will still exist for a period after the distraction behavior ends. Therefore, at least on highways or expressways, it is necessary to prohibit the use of smartphones to view all kinds of information.

In addition, the research background of this paper is based on the environment of highways or expressways. Natural driving research on urban roads is more complicated, and needs to be further studied.

## Figures and Tables

**Figure 1 ijerph-16-01464-f001:**
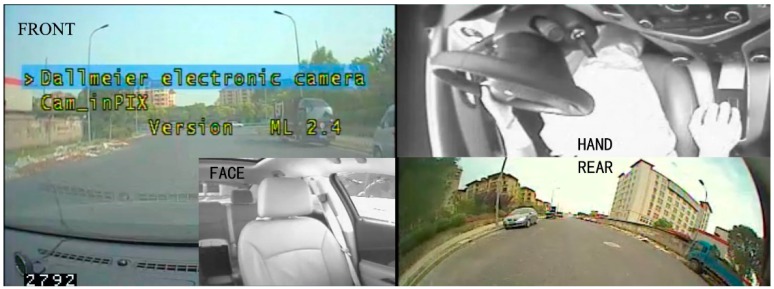
The framework of video data in SH-NDS (Shanghai Natural Driving Research Project).

**Figure 2 ijerph-16-01464-f002:**
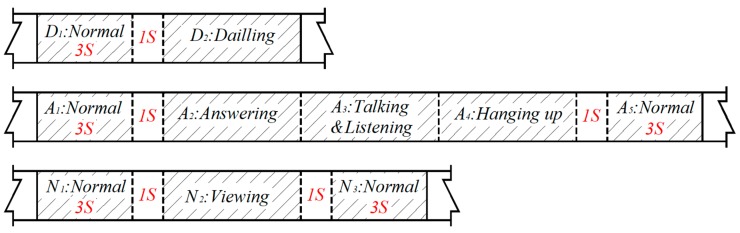
The area of data extraction.

**Figure 3 ijerph-16-01464-f003:**
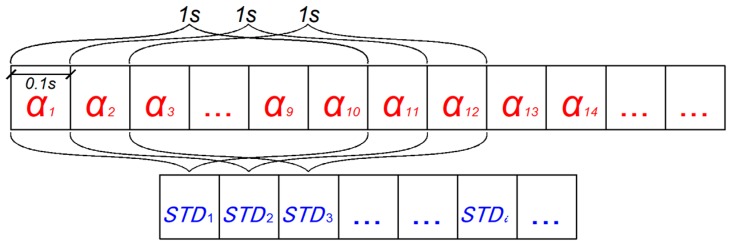
Moving time window.

**Figure 4 ijerph-16-01464-f004:**
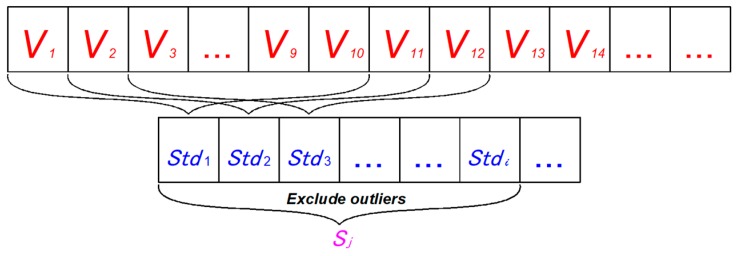
Calculation diagram of $S_{j}$.

**Figure 5 ijerph-16-01464-f005:**
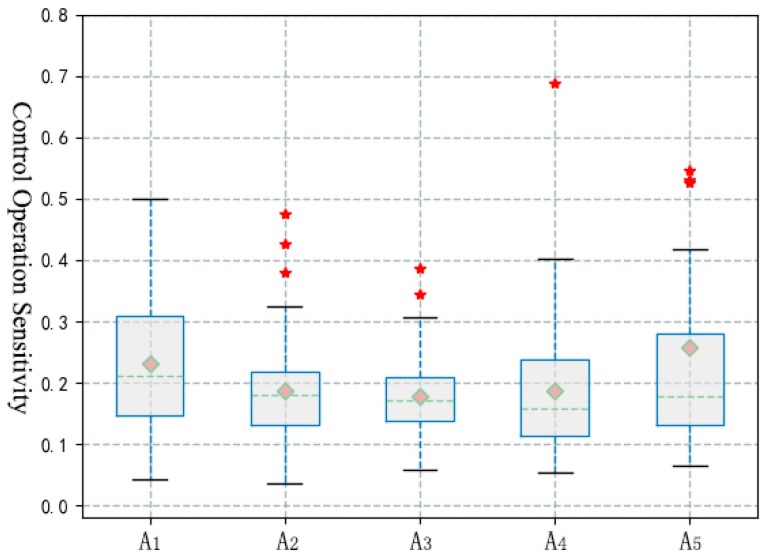
Statistical results of sensitivity index during answering calls.

**Figure 6 ijerph-16-01464-f006:**
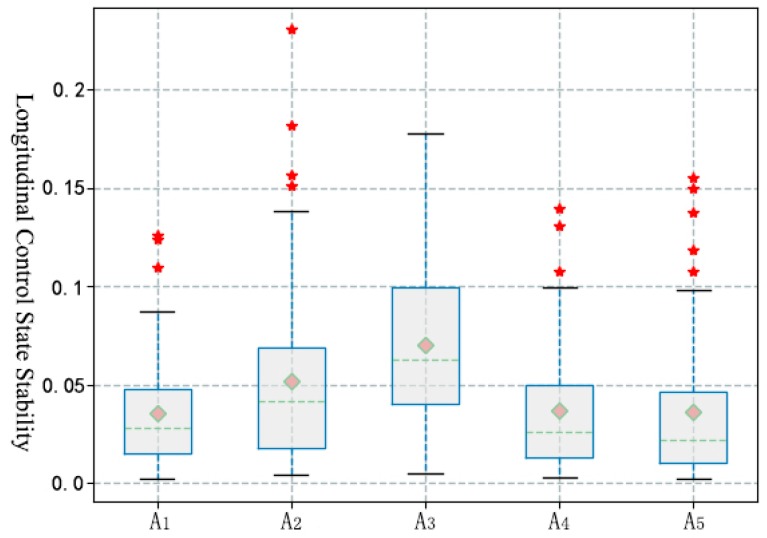
Statistical results of longitudinal stability index during answering calls.

**Figure 7 ijerph-16-01464-f007:**
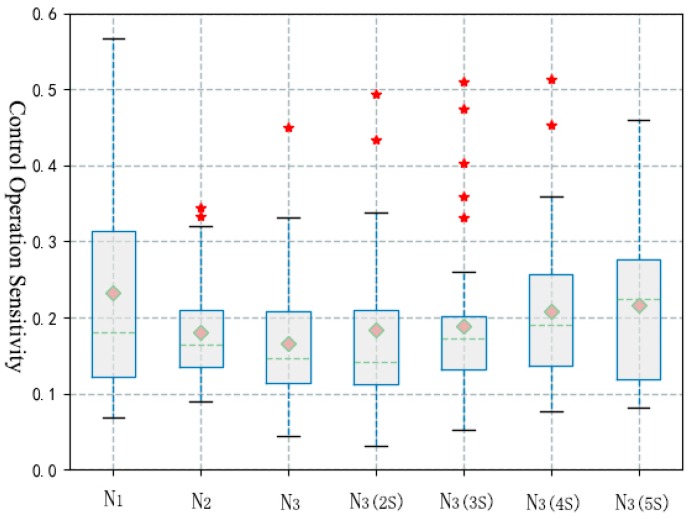
Statistical results of control operation sensitivity during viewing information.

**Table 1 ijerph-16-01464-t001:** Function introduction of data acquisition devices.

Data Acquisition Equipment	Introduction
Doppler radar	The equipment collects the relative distance and relative speed with surrounding vehicles. The measuring range is 40 m transversal and 150 m longitudinal. The acquisition frequency is 10 Hz.
Triaxial accelerometer	The acceleration and angular velocity of the vehicle in three directions are collected for determining the vehicle motion state, and the data acquisition frequency is 10 Hz.
Global Positioning System (GPS)	Vehicle coordinate information is collected, and data acquisition frequency is 1 Hz.
Multi-camera	A total of four cameras capture the driving behavior of the vehicle’s front view, rear view, driver’s face and hand.

**Table 2 ijerph-16-01464-t002:** Description of distraction operations.

Task	Sub-Operations	Description	Judging Criteria
Hand-held call	Answering	Check the incoming call with one hand and prepare the call.	Driver’s line of sight starts to move.
	Dialing	Click the screen with one hand to dial.	Driver’s line of sight starts to move.
	Talking and listening	Drive with one hand and communicate with a phone.	Put the phone at the ear.
	Hanging up	Stop talking and put the phone down with one hand.	Put the phone away from the ear.
View information	Viewing	Click on the screen with one hand to browse information. Keep eyes off the road for a long time.	Driver’s line of sight starts to move.

**Table 3 ijerph-16-01464-t003:** Sample description of each operation.

Task Category	Sample Size (Number of Events)	Sub-Operation	Average Duration (s)
Answer the call	60	Answering	5.5
		Talking and listening	50.1
		Hanging up	5.4
Dial the call	39	Dialing	21.5
View information	35	viewing	25

**Table 4 ijerph-16-01464-t004:** Results summary of main effects and comparisons during answering calls.

Comparison	Control Sensitivity	Control State
A_1_–A_2_	T(59) = 3.010, *p* = 0.004	T(59) = −2.484, *p* = 0.016
A_1_–A_3_	T(59) = 3.580, *p* = 0.001	T(59) = −5.751, *p <* 0.01
A_2_–A_3_	T(59) = 0.915, *p* = 0.364	T(59) = −2.752, *p* = 0.008
A_3_–A_4_	T(59) = −0.860, *p* = 0.393	T(59) = 6.446, *p* < 0.01
A_3_–A_5_	T(59) = −2.541, *p* = 0.014	T(59) = 5.420, *p* < 0.01
A_4_–A_5_	T(59) = −2.528, *p* = 0.014	T(59) = −0.165, *p* = 0.869

**Table 5 ijerph-16-01464-t005:** Results summary of control operation sensitivity during viewing information.

**Sensitivity Index**	**N_1_–N_2_**	**N_2_–N_3_**	**N_1_–N_3_**
	T(34) = 2.277 *p* = 0.029	T(34) = 1.259 *p* = 0.217	T(34) = 2.853 *p* = 0.007
**N_2_–N_3(2S)_**	**N_2_–N_3(3S)_**	**N_2_–N_3(4S)_**	**N_2_-–N_3(5S)_**
T(34) = −0.164 *p* = 0.871	T(34) = −0.493 *p* = 0.625	T(34) = −1.788 *p* = 0.083	T(34) = −2.279 *p* = 0.029

**Table 6 ijerph-16-01464-t006:** Results of Pearson correlation test.

		Duration	Control State
Duration	Pearson Correlation	1	−0.034
	*p*-value		0.760
Control state	Number of samples	81	81

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
