# Peer review of "Effect of Using Mobile Phones on Driver’s Control Behavior Based on Naturalistic Driving Data"

_ijerph, 2019, doi:10.3390/ijerph16081464_

Reviewer 1 Report

Overall Impression: This manuscript reports the impact of cell phone use while driving towards drivers’ control behaviors utilizing naturalistic driving study data. The results show strong correlation between distracted operations and driving control behavior. The approach of utilizing naturalistic driving data is innovative to account for environmental differences when one engages in distracted driving; however, some improvements to this paper are needed to emphasis the innovative nature of utilizing this data.

·       Spelling of behavior 

Introduction:

·       The authors discuss the importance of cognitive and visual distraction types when engaging in various secondary tasks, however, do not discuss the importance of manual distraction. The severity of taking ones’ hands off the steering wheel results in the consequences discussed such as headway distance, speed, veering off the road, etc. Additional information and references for manual distractions severity and consequences are needed because one of the main operations discussed is dialing a mobile phone.

Methods:

·       Page 4, line 1-7: May want to point out if all participants are driving automatic or manual cars. May differ for hand movements.

·       Page 4, line 13-14: Again, may want to include the manual distraction aspect of dialing a phone.

·       Page 4, line 22: Table 2: For dialing why was the judging criteria only driver’s line of sight starts to move. If there were cameras on hands, why wasn’t that used as well? Please discuss and justify.

Discussion:

·       The authors discuss the age restrictions for participants on Page 3, Line 10. You may want to include in the discussion the importance of including younger age ranges for future research. How distracted driving and hand-held vs. non hand-held devices may be different among 18- 26 year old.

·       How would the results have implications for policy?

Author Response

The comments are much appreciated and have helped in improving the quality of this paper. The responses are as follows.

Point 1: Spelling of behavior

Response 1: The error has been modified.

Point 2: The authors discuss the importance of cognitive and visual distraction types when engaging in various secondary tasks, however, do not discuss the importance of manual distraction. The severity of taking ones’ hands off the steering wheel results in the consequences discussed such as headway distance, speed, veering off the road, etc. Additional information and references for manual distractions severity and consequences are needed because one of the main operations discussed is dialing a mobile phone.

Response 2: Manual distraction should not be ignored when using the phone. In previous studies of using mobile phones during driving, it is generally believed that manual-visual distraction has a major impact. Relevant literatures and discussion have been supplemented to illustrate the severity of the effects of manual-visual distraction on driving.

... In addition, most operations of using mobile phones are manual-visual distractions, such as texting and dialing. And the effect of manual-visual distractions on driving performance is negative greater than just visual distraction [15-17] ... (Line 42, Page 1)

The purpose of this paragraph is to emphasize the seriousness of the influence of manual-visual operations in the process of using mobile phones, as well as the neglect of cognitive distraction in laws and regulations, so as to propose to analyze the difference of the influence of different operations in the process of using mobile phones.

Point 3: Page 4, line 1-7: May want to point out if all participants are driving automatic or manual cars. May differ for hand movements.

Response 3: The vehicles used in the experiment were all automatic cars provided by general motors. The type of vehicle will not be an interference factor. We added more details about the experiment vehicles.

“The experimental vehicles are all Chevrolet automatic transmission vehicles.” (Line 13, Page 3)

 Point 4: Page 4, line 13-14: Again, may want to include the manual distraction aspect of dialing a phone.

Response 4: The paper discusses the manual distraction in the process of using mobile phones.

“...Amongst these sub-operations, the degree of visual disturbance is reduced successively during viewing, dialing, answering and hanging up. In addition, answering, dialing and viewing create conspicuous manual distraction...” (Line 14, Page 4)

Point 5:   Page 4, line 22: Table 2: For dialing why was the judging criteria only driver’s line of sight starts to move. If there were cameras on hands, why wasn’t that used as well? Please discuss and justify.

Response 5: In the process of sample extraction, we judge the occurrence of distracted behavior by observing the driver's hand video to select the sample preliminary. When extracting the data, we need to identify specific moment when the driver's distraction starts or ends. And when the driver has the intention of distraction, the first thing that will happen is visual bias. Therefore, the deviation of the line of sight was taken as the criterion to ensure the accuracy of the sample data.

Point 6:   The authors discuss the age restrictions for participants on Page 3, Line 10. You may want to include in the discussion the importance of including younger age ranges for future research. How distracted driving and hand-held vs. non hand-held devices may be different among 18- 26 year old.

Response 6: Thanks for your comments. The influence of age on distracted driving is an important research direction. Previous studies have shown that younger people use their phones more frequently while driving, with more dramatic effects. Some countries have enacted relevant laws on this issue for young people. Considering the property protection of vehicles, the subjects of this experiment are mostly middle-aged people with long driving experience, stable income and high travel frequency. The focus of this study is on the differences in the effects of different behaviors. In future studies, we will attempt to expand the age range of the experimental group.

Point 7: How would the results have implications for policy?

Response 7: The paper supplements the discussion of policy from two aspects.

First, although the driver's observation time of the driving environment isn’t shortened during the talking and listening, the sensitivity of the driver's control operation and the stability of the speed control are significantly reduced, and the degree of influence is significantly higher than other distracting operations. Although the hand-free devices partially reduce some manual-visual operations, it doesn’t alleviate the cognitive distraction. Therefore, there should be restrictions on hand-free calling. We found this problem through this research, and subsequent research will be conducted on the behavior of hand-free calling.

“... In the process of hand-held calls, talking and listening leads to a decrease in the sensitivity of the driver's control operation, and the degree of influence is no different from other manual-visual operations. In addition, talking and listening leads to the most significant reduction in the stability of longitudinal control state in the process of answering calls. The influence of talking and listening operation on driver’s control behavior is no less than that of other operations, indicating that banning only the handheld communication mode cannot eliminate the security risks caused by the cognitive distraction. The use of hands-free calls should be restricted. Natural driving research for hands-free calling will be conducted in subsequent studies....” (Line 3, Page 11)

At present, China's road traffic safety law only prohibits the use of mobile phones to make and receive handheld calls and edit text messages while driving. With the development of smart phones, drivers are more and more frequently checking social messages, maps through their mobile phones. The emergence of ride-hailing has led many drivers to rely on their mobile phones for daily driving. The results of this paper show that the influence of this kind of behavior on driving control is significant, and the influence time is longer, indicating that this kind of behavior should be restrained.

“...Nowadays, with the development of the functions of smart phones, the behavior of using mobile phones to edit short messages is less, while the behavior of browsing and viewing is more frequent. The advent of social media, mobile navigation and especially ride-hailing services has left drivers completely dependent on their phones. However, with the proliferation of this behavior, there is a lack of effective regulation in China. It can be seen that the viewing behavior not only has a significant influence on the sensitivity and stability of driving control, but also the influence will still exist for a period after the distraction behavior ends. Therefore, at least on highways or expressways, it is necessary to prohibit the use of smartphones to view all kinds of information...” (Line 11, Page 11)

Because the background of this study is highway or expressway, and the initial speed of sample is high. Therefore, the impact caused by the change of control behavior may be greater. So, the relevant policies should be applied in the same environment.

Reviewer 2 Report

I think this is a nice paper. I am glad to see some data from China using a naturalistic methodology. These are my comments: - Line 2 Pag 12: Authors mention "stress". However, this is not theoretically accurate. The resource allocation should be explained due to a mismatch between driving demands and driver capability. This doesn't mean that drivers are stressed. Driving demands are typically seen as workload or feeling of risk. Check the Driver behaviour theory of Fuller.  Please fix the language in the manuscript. - Authors mentioned: "driver tends to weaken the longitudinal control activities". Some researchers argue that this could be that they are overcorrecting their lane position, which is positive. Any thoughts on this? We need to avoid suggesting that a behaviour change is harmful when it is not necessarily true. Has this been linked to crash risk? Please check some research/reviews on this area. - The discussion of self-regulatory behaviours was insufficient. This was one of the main findings.  Please include references to support this. Examples: Oviedo-Trespalacios, O., Haque, M. M., King, M., & Demmel, S. (2018). Driving behaviour while self-regulating mobile phone interactions: A human-machine system approach. Accident Analysis & Prevention, 118, 253-262. Young, K. L., Osborne, R., Koppel, S., Charlton, J. L., Grzebieta, R., Williamson, A., ... & Senserrick, T. (2019). What are Australian drivers doing behind the wheel? An overview of secondary task data from the Australian Naturalistic Driving Study. Journal of the Australasian College of Road Safety, 30(1), 27-33.  - Please discuss how this manuscript contributes to the knowledge in the area. What is new and specific to China? Any insights for policy? Not much of the information presented here is new.

Author Response

The comments are much appreciated and have helped in improving the quality of this paper. The responses are as follows.

Point 1: Line 2 Page 11: Authors mention "stress". However, this is not theoretically accurate. The resource allocation should be explained due to a mismatch between driving demands and driver capability. This doesn't mean that drivers are stressed. Driving demands are typically seen as workload or feeling of risk. Check the Driver behaviour theory of Fuller. Please fix the language in the manuscript.

Response 1: Thanks for your comments, and the word “stress” isn’t accurate. The related expression has been replaced with "workload" to describe the effect of distracting behavior.

Point 2: Authors mentioned: "driver tends to weaken the longitudinal control activities". Some researchers argue that this could be that they are overcorrecting their lane position, which is positive. Any thoughts on this? We need to avoid suggesting that a behaviour change is harmful when it is not necessarily true. Has this been linked to crash risk? Please check some research/reviews on this area.

Response 2: Thanks for your comments. "driver tends to weaken the longitudinal control activities" is inaccurate. We have made changes to the relevant narrative. The driver's tendency to reduce the intensity of longitudinal control activity may also be due to the reduction of driving demand by drivers, which help to reduce the risk. Relevant literatures and discussion have been added.

“... Some studies show that distracted drivers often exhibit self-regulatory behaviors, such as reducing driving speed, increasing the distance to the lead vehicle, and less lane changes, which are considered can reduce the driving demand and accident risk [34-36]. Some studies have also found less lane deviation during hand-held calls and increased lane deviation during texting [37-40], which is consistent with the results of this paper. And also, some studies report don’t find similar results [41,42], especially in natural driving studies [31,32]. This may be related to the complexity of natural driving samples. At the beginning of this study, some interfering factors are eliminated by setting sample screening conditions and index construction. The intensity of the control activity can reflect the decision-making willingness of drivers. The results show that drivers are unable to devote more attention resources in the face of extra workload caused by complex manual-visual distractions, and tend to reduce the longitudinal control activities, so as to ensure that the intensity of lateral control activity is not affected. As the distraction intensifies, the strategy shifts....” (Line 11, Page 9)

And some studies have shown that these behaviors can have a negative impact on nearby vehicles. The effect of such behavior on driving risk remains unknown when the vehicle is on a highway or expressway.

“... Although this behavior is a kind of self-protection of the driver, the longitudinal self-regulatory behaviors of drivers have a significant negative impact on the surrounding traffic flow [43]. Under the condition of high-speed driving, the influence of this regulation on driving risk in the region needs to be further explored.....” (Line 22, Page 9)

Point 3: The discussion of self-regulatory behaviours was insufficient. This was one of the main findings.  Please include references to support this. Examples: Oviedo-Trespalacios, O., Haque, M. M., King, M., & Demmel, S. (2018). Driving behaviour while self-regulating mobile phone interactions: A human-machine system approach. Accident Analysis & Prevention, 118, 253-262. Young, K. L., Osborne, R., Koppel, S., Charlton, J. L., Grzebieta, R., Williamson, A., ... & Senserrick, T. (2019). What are Australian drivers doing behind the wheel? An overview of secondary task data from the Australian Naturalistic Driving Study. Journal of the Australasian College of Road Safety, 30(1), 27-33.

Response 3: There is evidence that drivers try to reduce the driving task demand by an adjustment in driving behavior. Self-regulatory behavior under natural driving conditions plays an important role in avoiding driving risks. We supplemented some literatures and discussion of self-regulatory behavior.

... Some studies show that distracted drivers often exhibit self-regulatory behaviors, such as reducing driving speed, increasing the distance to the lead vehicle, and less lane changes, which are considered can reduce the driving demand and accident risk [34-36]. Some studies have also found less lane deviation during hand-held calls and increased lane deviation during texting [37-40], which is consistent with the results of this paper. And also, some studies report don’t find similar results [41,42] ...” (Line 11, Page 9)

This finding is aslo one of the main conclusions of this paper.

“…The study finds that drivers used different resource allocation strategies to deal with the extra workload caused by distraction under different distraction operations. Typical manual-visual distraction operations, such as answering, dialing and viewing, can result in a significant interference in the driver's attention resources. Therefore, the driver has adopted some coping strategies under different distraction intensity. The driver tends to decrease the longitudinal control activities in the process of answering or dialing, and decrease the lateral control activities in the process of viewing, which plays an important role in reducing driving demand.…”

Point 4: Please discuss how this manuscript contributes to the knowledge in the area. What is new and specific to China? Any insights for policy? Not much of the information presented here is new.

Response 7: The paper supplements the discussion of policy from two aspects.

First, although the driver's observation time of the driving environment isn’t shortened during the talking and listening, the sensitivity of the driver's control operation and the stability of the speed control are significantly reduced, and the degree of influence is significantly higher than other distracting operations. Although the hand-free devices partially reduce some manual-visual operations, it doesn’t alleviate the cognitive distraction. Therefore, there should be restrictions on hand-free calling. We found this problem through this research, and subsequent research will be conducted on the behavior of hand-free calling.

“... In the process of hand-held calls, talking and listening leads to a decrease in the sensitivity of the driver's control operation, and the degree of influence is no different from other manual-visual operations. In addition, talking and listening leads to the most significant reduction in the stability of longitudinal control state in the process of answering calls. The influence of talking and listening operation on driver’s control behavior is no less than that of other operations, indicating that banning only the handheld communication mode cannot eliminate the security risks caused by the cognitive distraction. The use of hands-free calls should be restricted. Natural driving research for hands-free calling will be conducted in subsequent studies....” (Line 3, Page 11)

At present, China's road traffic safety law only prohibits the use of mobile phones to make and receive handheld calls and edit text messages while driving. With the development of smart phones, drivers are more and more frequently checking social messages, maps through their mobile phones. The emergence of ride-hailing has led many drivers to rely on their mobile phones for daily driving. The results of this paper show that the influence of this kind of behavior on driving control is significant, and the influence time is longer, indicating that this kind of behavior should be restrained.

“...Nowadays, with the development of the functions of smart phones, the behavior of using mobile phones to edit short messages is less, while the behavior of browsing and viewing is more frequent. The advent of social media, mobile navigation and especially ride-hailing services has left drivers completely dependent on their phones. However, with the proliferation of this behavior, there is a lack of effective regulation in China. It can be seen that the viewing behavior not only has a significant influence on the sensitivity and stability of driving control, but also the influence will still exist for a period after the distraction behavior ends. Therefore, at least on highways or expressways, it is necessary to prohibit the use of smartphones to view all kinds of information...” (Line 11, Page 11)

Because the background of this study is highway or expressway, and the initial speed of sample is high. Therefore, the impact caused by the change of control behavior may be greater. So, the relevant policies should be applied in the same environment.

Reviewer 3 Report

The authors presented the study of using mobile phone in relation to driving behavior using 2 months driving dataset which is actually quite interesting. The manuscript is well-written and presentation is well-organized.

Here are some comments.

1) Please verify the x-axis value in Fig. 5.

2) Is there any questionnaire involved? I think a further comparison with drivers' own sense of driving behavior with the analyzed driving behavior. Some "good" driving skill driver might feel those are not distraction and cause minor problem to them with their long experiences of driving, but might be an issue to others. If applicable, including this result could increase the novelty of the study.

3) Are the subject drivers gender different? Is there any major different between male and female driving behavior? 

Author Response

The comments are much appreciated and have helped in improving the quality of this paper. The responses are as follows.

Point 1: Please verify the x-axis value in Fig. 5

Response 1: It is our oversight and the error has been modified.

Point 2: Is there any questionnaire involved? I think a further comparison with drivers' own sense of driving behavior with the analyzed driving behavior. Some "good" driving skill driver might feel those are not distraction and cause minor problem to them with their long experiences of driving, but might be an issue to others. If applicable, including this result could increase the novelty of the study.

Response 2: Thanks for your comments. It's a great research direction. Make a comparative analysis between drivers' self-cognition and actual driving performance, or analyze the differences in driving performance of drivers with different self-cognition. This will give a deeper understanding of distracted driving. It is a pity that we did not conduct a questionnaire survey on the participants. In addition, the diversity of the samples is affected because the subjects are all drivers who have been driving for more than five years. The focus of this study is mainly on the differences in the effects of different behaviors. We will conduct supplementary investigations and experiment in further studies.

Point 3: Are the subject drivers gender different? Is there any major different between male and female driving behavior?

Response 3: Thanks for your comments. The participants in this experiment are mainly male drivers, which does have some limitations. Some studies suggest that gender influences the frequency of distracted driving and gender has less effect on driving performance when distracted. However, gender and age are important factors in related studies. Considering that the focus of this study does not involve relevant content, it will be supplemented in the following natural driving experiment.

Round  2

Reviewer 3 Report

Authors had revise the comments correctly. Recommend for acceptance in current form.